# Digitalization of Calibration Data Management in Pharmaceutical Industry Using a Multitenant Platform

**Tuukka Mustapää [1,*], Juho Nummiluikki [2] and Raine Viitala [1]**

[1] Department of Mechanical Engineering, School of Engineering, Aalto University, 02150 Espoo, Finland; raine.viitala@aalto.fi
[2] Beamex Oy Ab, 68600 Pietarsaari, Finland; juho.nummiluikki@beamex.com
* Correspondence: tuukka.mustapaa@aalto.fi

**Abstract:** The global quality infrastructure (QI) has been established and is maintained to ensure the safety of products and services for their users. One of the cornerstones of the QI is metrology, i.e., the science of measurement, as the quality management systems commonly rely on measurements for evaluating quality. For this reason, the calibration procedures and the management of the data related to them are of the utmost importance for the quality management in the process industry and given a high priority by the regulatory authorities. To overcome the relatively low level of digitalization in metrology, machine-interpretable data formats such as digital calibration certificates (DCC) are being developed. In this paper, we analyze the current calibration processes in the pharmaceutical industry, and the requirements defined for them in the relevant standards and regulations. For digitalizing the calibration-related data exchange, a multitenant cloud platform-based method is presented. To test and validate the approach, a proof of concept (POC) implementation of the platform is developed with a focus on ease and cost-efficiency of deployment and use while ensuring the preservation of traceability and data integrity. The POC is based on two industrial use cases involving organizations with different roles in the metrology infrastructure. In the testing, the presented approach proves to be an efficient method for organizing the calibration data exchange in industrial use.

**Keywords:** digitalization; quality management; pharmaceutical industry; digital calibration certificate; metrology; traceability; platform

## 1. Introduction

Whenever a product or service is brought to market, it is crucial that the product or service in question is safe for the consumers and the environment. For these purposes, a global quality infrastructure (QI) consisting of both public and private organizations has been established [1]. The cornerstones of the QI are metrology, standardization, accreditation, conformity assessment, and market surveillance, which have their own sub-infrastructures, such as the metrology infrastructure (MI), which all have their own international organizations [2]. The global cooperation of the QI is coordinated by the International Network on Quality Infrastructure (INetQI) [3]. Due to the differences in the roles of the organizations and regulatory frameworks in the QI, there are organizations and regulatory bodies covering the different parts of the QI on national or regional levels. Due to this divisioning, there are possibilities for overlapping or conflicting regulations, which the regulatory bodies ideally aim to avoid. Currently, as significant efforts are being made to digitalize the QI [1,4–7], the differences in current practices and requirements quickly become apparent, causing challenges in the implementation of digital solutions. For ensuring that the data formats and digital processes meet their respective requirements and are applicable globally in different applications, harmonization is needed as the current requirements may vary by the domain or region. Otherwise, it could be impossible or at least highly cost-inefficient for the organizations participating in the maintaining of

the infrastructures, such as calibration laboratories and service providers, to adapt their services to be interoperable with the varying systems used by individual customers.

In this paper, we focus on the industrial part of the MI, in which the determining of the measurement uncertainty of individual instruments and traceability of measurements to the measurement unit system are established through calibrations [8]. As calibrations are a confirmative part of the quality management of processes that are based on measurements, they are not directly profitable operations but instead are necessary for avoiding prohibitive expenses and delays or fulfilling compliance requirements, allowing access to markets. For this reason, estimating the total economic benefits of the digital transformation of the MI for all the involved organizations individually is complex. The same also applies to other similar processes in different parts of the QI. As the purpose of the QI, and thus also the MI, is mainly to oversee and support the industry in providing reliable and safe products and services to customers [1], the most economical benefits from the digital transformation are formed in the customer end of the infrastructure. In the MI, this includes, e.g., manufacturing companies that can have several thousands of measurement instruments monitoring and controlling production lines and processes. For these companies, a significant part of the value of the digitalization comes from the improvements in efficiency and reduced need for manual work through automation, which leads to savings from reduced human resources tied to quality management. In the case of calibrations, where the costs and possible savings from managing the information of the instruments are relative to the number of the instruments that need to be regularly calibrated, these benefits will not be as significant for smaller service providers. This means that any investments in the digital transformation become difficult to justify without support and demand from the customers.

A good example of an industry that is reliant on the MI is the pharmaceutical industry, where measurements provide the means for controlling the drug manufacturing processes. Thus, the quality and safety of the pharmaceutical products are directly dependent on the reliability of the process measurements. For this reason, ensuring the trustworthiness of the process instruments is an essential part of quality management in the pharmaceutical industry, which is why the measurement systems and their maintenance and calibration procedures are highly regulated [9–11]. Ensuring the data integrity of the measurements and any calibration-related data in particular is of the utmost importance for this purpose, which is why detailed records and audit trails for the processes are required [12]. Consequently, any processes including human operations, e.g., related to the documentation and handling of calibration data, will typically require several inspection and approval procedures to prevent the occurrence of human errors.

Since harmonization of the data formats is essential for the interoperability and overall efficiency of digital systems, important areas requiring in-depth examination are the established standards, regulations, and guidelines that provide the framework for the current data formats and processes. Thus, a key question for the success of the digitalization efforts becomes how compatible and applicable are the requirements defined for the different parties in QI in a fully digital environment.

In general, the success of industry-wide transition in the digital environment relies on organizational capabilities to adapt to and uptake new technologies and systems when necessary. Thus, an important aspect of the digital transformation is ensuring the ease of adaptation, inexpensiveness, and sufficient scalability of the processes and systems so that the requirements of various types and sizes of organizations can be fulfilled. One possible solution for arranging these kinds of systems and services in a scalable manner is the utilization of a common cloud platform. An example of such a concept in the domain of metrology is the European Metrology Cloud project, aiming to advance the digitalization of legal metrology in Europe [7,13].

Optimal digitalization of the calibration data management requires a thorough understanding of the underlying processes and workflows. In this paper, we investigate the current practices and general requirements for calibration data management as a part

of the quality management in the pharmaceutical industry. We analyze the significance and feasibility of the requirements for the harmonization of digital processes. The paper presents ways to optimize calibration data management processes, allowing improved efficiency by reduced manual work and removing the possibility of human errors in calibration data management. Thus, the traceability and data integrity can be preserved between the organizations in the calibration chains. Furthermore, the proposed approach enables data analysis of the calibration data in a completely different scale compared to the printed A4/paper on glass (e.g., PDF) approach. A proof of concept (POC) of the method was developed by prioritizing the ease of adaptation to the digital processes. Thus, the implementation utilized systems already widely used in the industry, e.g., for user and access management. Thus, the set-up and operation of the platform required a less complex infrastructure compared to a fully proprietary infrastructure, as used in [13,14].

To summarize, we present the following contributions beyond the state-of-the-art:

1. Investigation of calibration procedures in the digital calibration certificate (DCC) [15] context, including the core standards and regulations in the case of the pharmaceutical industry;
2. Analysis of the applicability of the standards and regulations for the implementation of machine-proof data formats and application programming interfaces (APIs), acknowledging that the existing procedures, standards, and regulations have been prepared for human interpretation;
3. Optimized digital data management processes for preserving traceability and data integrity in calibration chains;
4. A concept for a multitenant platform for establishing ecosystems for collaborating organizations within the metrology infrastructure;
5. A proof of concept (POC) realization of points 3 and 4.

The paper is organized as follows: Section 2 provides the relevant background on the digitalization of the metrology infrastructure, current developments in the use of IoT in manufacturing and quality management, and requirements for the calibration data management in the process and pharmaceutical industries. The harmonization of the digitalized calibration data management based on the requirements for different organizations is discussed in Section 3, and the proposed approach for the exchange of digital calibration data within the metrology infrastructure is presented in Section 4. The possibilities of the digital calibration data management, remaining challenges, and proposed research topics are discussed in Section 5. Section 6 concludes the paper.

## 2. Background

### 2.1. Metrology Infrastructure as a Part of the Quality Infrastructure

The worldwide MI is built upon standards, mutual trust, and recognition between organizations operating around the world [16]. To make the infrastructure as comprehensive as possible, it consists of organizations with different hierarchical roles. At the top of the hierarchy are the national metrology institutes (NMIs) and designated institutes (DI), which maintain the SI system and metrological standards jointly defined by the International Bureau of Weights and Measures (BIPM). Figure 1 presents an illustration of how the MI is established as a part of the QI. The hierarchy of the MI is often depicted in the form of a triangle or pyramid as the amount of measurement instruments and references increases when moving from the NMIs towards the industrial measurement application. A few examples of how the MI has been implemented nationally in European countries are given in [17].

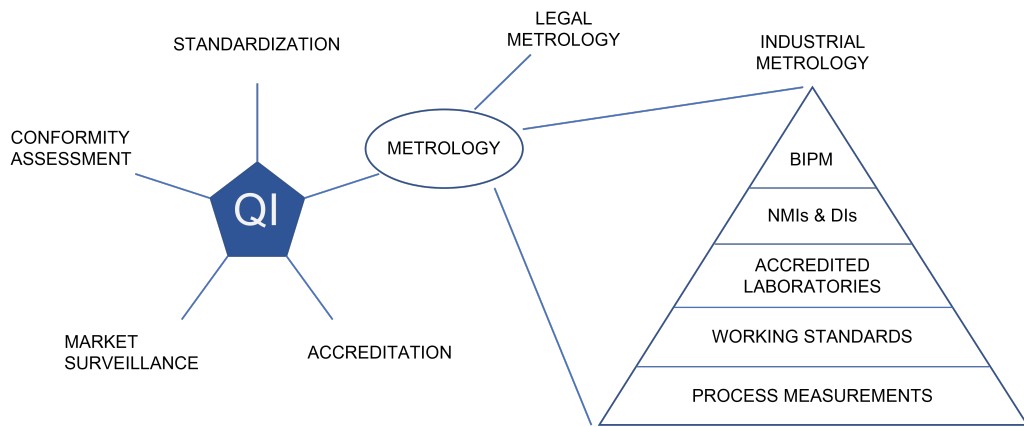

**Figure 1.** Metrology infrastructure as a part of the quality infrastructure (QI).

The measurement standards maintained by the NMIs or DIs are referred to as primary standards as their purpose is to provide as accurate and precise physical representations of the unit definitions as possible. Between the industrial applications and primary standards are secondary measurement standards, which are maintained by accredited laboratories and calibrated by the NMIs or DIs. Calibrations are essential to maintain the traceability to SI unit definitions, which ensures the comparability and trustworthiness of measurement results given by individual instruments [8]. Depending on the applications, additional levels of references may also be used in the industry. For example, in the pharmaceutical industry, where the number of instruments used for monitoring and controlling the manufacturing processes can be several thousand per manufacturing site, the use of working standards and travelling standards is common as they provide efficiency in terms of time and costs, e.g., by allowing the calibrations of the process instruments to be carried out on-site. Figure 2 shows an example of a calibration chain from the primary standard of an NMI all the way through to a process instrument of a pharmaceutical manufacturing company.

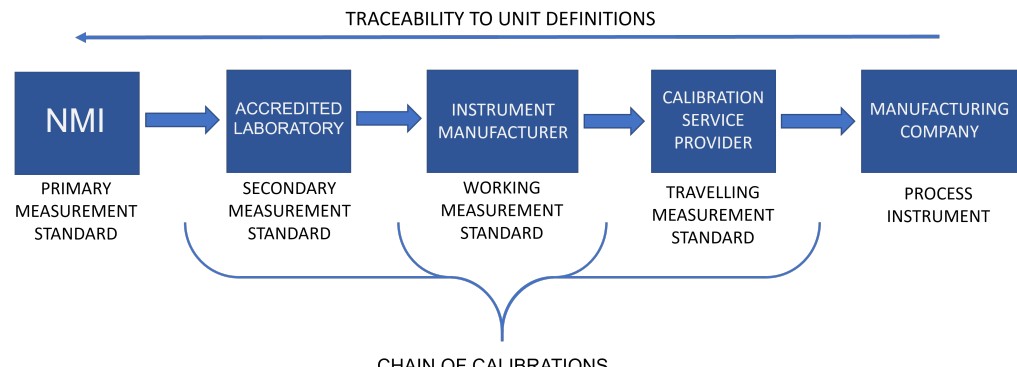

**Figure 2.** An illustration of traceability in a calibration chain.

Due to the complexity of the MI and diversity of the organizations involved, the progress of digitalization has been slow and in many cases data are still being managed in paper or paper-on-glass formats requiring human interpretation, e.g., in the case for the documenting of calibrations in calibration certificates [15]. For field calibrations, digitalized solutions have become more common as the costs of calibrations are proportional to the efficiency of the processes. However, these solutions tend to be instrument- and system-provider-specific, and due to competition, the willingness of the system providers to collaborate by harmonizing their systems and making them more open has been low. In addition, it is common that larger industrial companies cooperate with several instrument manufacturers and service providers. Consequently, in the worst case for the instrument owner, the data exchange from different organizations has been scattered in several sys-

tems and using different formats. Because of this, several human resources relative to the number of annual calibrations have been tied up in the calibration management alone to ensure compliance with the quality management regulation, meaning that it has been a significant expense.

As the digitalization in the industry and other sectors is progressing and the collection and use of data is growing rapidly, the topics concerning data quality and trustworthiness become more and more important, which is driving the research and development efforts in digitalization within the metrology community [7,18]. The ongoing efforts include the definition and development of digital formats for presenting metrological information, e.g., data models for presenting metrologically relevant data as semantic metadata [19–21] or formats for digital calibration certificates [15,22–24]. For this work, the exchange of DCCs was tested using the DCC format originally defined by PTB and further developed in EMPIR SmartCom and Gemimeg research projects [4,5,25–28].

Digitization of the data formats sets new requirements for the development of the infrastructure as its operation has been based on mutual recognition and trust between organizations [16]. Transition to a digital environment means that also this trust needs to be established digitally to enable its representation in a machine-understandable format [29–31]. For this purpose, data security and cryptographical solutions, e.g., the use of digital signatures, are relevant [15,17].

### 2.2. IoT in Pharmaceutical and Process Industries

The digital transformation of the manufacturing industry is typically referred to as Industry 4.0 based on the significance and potential of digitalization in industrial settings. The key technologies of Industry 4.0 include smart IoT devices and sensors that enable the collection of vast amounts of data from different manufacturing processes and operations. Combined with the technological advancements in computing power, analyzing methods for big data and machine learning, Industry 4.0 enables significant improvements in the efficiency of manufacturing processes [32].

In addition to automation, digitalization also provides new possibilities to enhance the work where automation is not feasible or needs to be supported by manual operations, i.e., smart working. Smart-working-enabling technologies include, e.g., wearable IoT devices and augmented reality, which aid the interaction between the operators and manufacturing systems [33]. IoT technologies can also be used to improve worker safety in the working environment [34].

One of the main technologies that is being studied and developed based on the Industry-4.0-enabling technologies is the cyber-physical systems, i.e., digital twins, in which, e.g., data, simulation models, and predictive analytics are used to form as accurate a digital representation of a physical entity as possible to further improve the possibilities to analyze their performance and behavior [35–38].

In terms of quality management, the effects of Industry 4.0 are prominent in the digitalization of methodologies such as total quality management (TQM) and the use of IoT technologies in quality management operations and processes [39]. In some cases, the advancements in IoT technologies have also led to situations where the traditional quality management is not in alignment with the requirements of the IoT [40].

In the process and pharmaceutical industries specifically, the benefits of IoT and Industry 4.0 can be seen as being the same as in other manufacturing industries [41,42]. However, in the pharmaceutical industry, digitalization has been slowed due to the interdependence of processes and strict regulations requiring, e.g., comprehensive validation of newly developed systems and software, due to which the investments in Industry 4.0 solutions can be both risky and expensive [43–45]. In a survey by Reinhardt et al. the main areas of focus for Industry 4.0 adoption in the pharmaceutical industry were process optimization, production performance monitoring, insurance of regulatory compliance, and production downtime minimization [46].

*2.3. Requirements for the Calibration Data Management and Data Formats*

The Section 7.6 of the ISO 9001 standard for quality management systems contains the general requirements for the handling of measuring and monitoring equipment [47]. The standard requires that the equipment used for quantitative measurements must be calibrated periodically, for which the instrument owner must define an operation procedure including documentation.

For the process and pharmaceutical industry, there are legislative requirements for the federal legislations, e.g., pharmacopeias and good practices (GXP) such as good manufacturing practices (GMP) and good laboratory practices (GLP).

The main standard defining the requirements for the calibration data management and certificate formats for accredited laboratories is the ISO 17025 standard on general requirements for the competence of testing and calibration laboratories [48]. This standard, accompanied by other more general standards such as the ISO 9001, is used as the basis for the accreditation of calibration laboratories [49]. The requirements for the contents of a calibration certificate are defined in Section 7.8 of the ISO 17025 standard.

### 2.3.1. General Guidelines and Good Practices

In addition to international standards and regional legislations, there also exist several non-obligatory recommendations in the form of good practices or guidelines by different organizations, which have become widely used and unofficially agreed de-facto standards for the industry. These de-facto standards can be guidelines by NMIs or regional metrology organizations (RMO) on calibration procedures for specific instrument types, guidelines by societies and associations of the experts in the field [50,51], or guides and whitepapers by companies, e.g., instrument manufacturers, which are typically based on the relevant legislative requirements.

### 2.3.2. Mass as an Example Measurand

The differences that the standards, regulations, and guidelines of the different organizations may have can be pointed out by examining a singular physical quantity and the respective measurement methodologies. Harmonization is beneficial as differences in the calibration procedures can result in differences, e.g., in uncertainty evaluations [52]. Good examples of the harmonization of practices on a global level are the most recent revisions of the regulations and guidelines for mass and weighing, e.g., non-automatic weighing instruments.

Standards covering the use of non-automatic weighing instruments include the European EN 45501:2015 "Metrological aspects of non-automatic weighing instruments" [53] and American NIST Handbook 44 "Specifications, Tolerances, and Other Technical Requirements for Weighing and Measuring Devices" [54].

Regional guidelines for the calibration procedures for weighing instruments are published by the RMOs. In the case of non-automatic weighing instruments, these guidelines include, for example, Euramet Calibration Guideline No. 18 "Guidelines on the Calibration of Non-Automatic Weighing Instruments" [55], ASTM E898 "Standard practice for calibration of non-automatic weighing instruments" [56], SIM MWG7 "Guidelines on the calibration of non-automatic weighing instruments" [57], and JJF 1847 "Calibration Specifications for Electronic Balances" [58]. Harmonization of these guidelines has been based on the other organizations adapting their guidelines to commensurate with the EURAMET CG-18, which has also become better known in the Asian region, allowing the requirements to become a global de-facto standard [11].

On the industrial level, the requirements for the weighing instruments and other measurement instruments are defined in the regional or national regulations for pharmaceutical products. In the US Pharmacopeia, the requirements for balances are covered in General Chapter 41 "Balances" [9], with supporting guidance given General Chapter 1251 "Weighing on an Analytical Balance" [59]. Respectively, in the European Pharmacopeia, the requirements for balances are covered in General Chapter 2.1.7 "Balances for analytical

purposes" [10]. After the latest revision of the European Pharmacopoeia, the requirements are commensurate with the US Pharmacopeia, which has been a welcome advancement towards the harmonization of regulations [11].

2.3.3. Requirements for Data Integrity

One of the topics where improvements are pursued by investing in digitalization in the pharmaceutical industry is data integrity. The basis for the data integrity regulation compliance is typically referred to as the ALCOA+ principle [60,61], the definition of which is presented in Table 1.

**Table 1.** The definition of the ALCOA+ principle [60,61].

| Symbol | Meaning | Definition |
|---|---|---|
| A | Attributable | It must be clear when an action was taken and by whom, where the data came from, e.g., the systems, devices, or instruments used. |
| L | Legible and intelligible | The data and records must be comprehensible regardless of the record types. |
| C | Contemporaneous | Recordings must take place at the time of the action or observation and a timestamp must be included. |
| O | Original | If a record is copied, the original record must be distinguishable, and the copies must be verified as "true copies". Especially for electronic records, all relevant metadata must also be considered. |
| A | Accurate | All instruments used for collecting data must be accurate and controlled, e.g., calibrated, and any corrections or changes that need to be made in the records must be documented as amendments. |
| + | Available | A record must be accessible and available for audits over its lifetime. |
| | Complete | All relevant data and metadata must be included in the records and must not be omitted or deleted. |
| | Consistent | The timestamps in the records of actions and observations must be consistent with order of the steps in the operation procedure and be based on a common time reference. |
| | Enduring | The media used to store the records must be authorized and robust to ensure the preservation of the record over its lifetime. |

Regulative requirements for data integrity in manufacturing are either defined as a part of the GMPs and GLPs or in separate regulations such as the US Food and Drug Administration (FDA) 21 Code of Federal Regulations (CFR) Title 21 Part 11 [62] or Medicines & Healthcare Products Regulatory Agency (MHRA) Data Integrity Guidance and Definitions [63]. An example of a de-facto standard on data integrity is the ISPE GAMP Guide: Records & Data Integrity [12], where data integrity requirements defined in the regulations are approached in a more practical way.

Data integrity is also covered, although to a lesser extent, in the ISO 17025 Section 7.11, where requirements for the data management systems used by calibration laboratories for collecting, processing, recording, reporting storing, or retrieving data [48] are defined. However, as the requirements for the calibration certificates focus on the content that must

be included, regardless of whether the certificate is a physical or electronic document, the standard does not specify detailed requirements or methods for securing a digital file.

2.3.4. Harmonization Challenges for Developing Digital Solutions

The challenge in developing harmonized and interoperable formats and data exchange between the calibration service providers and the customers is that the requirements for the process and its documentation depend on the roles of the organizations in a calibration chain. For the national metrology institutes (NMI) and accredited laboratories, the calibration practices are mostly defined in the ISO 17025 standard and other requirements that are mandated as a part of the accreditation.

However, a significant amount of the calibrations performed in the industry are not done according to the ISO 17025, mostly due to costs. This includes field calibrations that are performed on-site, as this is far more cost-efficient in comparison to having all the process-controlling instruments calibrated at a separate laboratory. For these calibrations, the requirements are defined in the quality management regulations and legislations of the national and regional authorities. Compared to the ISO 17025, which specifically focuses on calibrations, the legislative requirements focus on the quality management system as a whole. The main principles regarding the calibrations of the instruments are no different, but the context and reasoning behind them is.

For NMIs and accredited laboratories, calibrations are a service that they provide for their customers, and they are regulated and audited to ensure that their processes are up to the level that is required, and this is why the requirements are very specific. On the other hand, in the industry's end of the calibration chain, the regulations are not as specifically defined in all of the cases. For example, instead of requiring, e.g., calibration processes to follow a specifically defined procedure, the companies are mandated to have written instructions for the processes, meaning that some of the details are left open for the companies to define themselves. Due to this openness, there are some guidelines and good practices that have been adopted as de-facto standards for calibration procedures. To shortly summarize the differences between the regulations for accredited calibrations and field calibrations, accredited calibrations are focused on an individual instrument whereas, in the field calibrations, the instrument is mainly considered as a part of a process line.

The effects of these differences in the requirements can be noticed when comparing a typical format for a field calibration certificate to a certificate issued by an accredited laboratory. The field calibration certificates must include information such as the identification of the position or location where the instrument has been used in a particular process line, which is relevant information for the quality assurance of that process line, and a calibration due date based on the calibration interval specified in the manufacturing company's requirements.

For the calibration data, the main difference is that the calibration data have been presented in a different way. The standards for the accredited laboratories require that the calibration results are presented including the measurement uncertainties of the measurement points, whereas in the regulations for the industry, it is relatively common that the deviations or measurement errors are required. Another example of these kinds of differences includes requirements for the use of different units in different parts of the world.

In addition to the variances in requirements, some challenges for the digitalization of the processes arise from the way in which the requirements have been defined in the standards and legislations, as the current formats are written for human interpretation. The International Electrotechnical Commission (IEC) has presented a utility model for SMART Standards [64]. The model defines the following levels for the digitization degree of a standard:

- Level 0: Paper format. Not suitable for direct automatic processing or usage.
- Level 1: Digital format (e.g., PDF) allowing automatic management and display of the document.

- Level 2: Machine-readable format. The structure of the document can be digitized and certain granular content can be exported (chapters, graphics, definitions, etc.). Content and presentation are separated.
- Level 3: Machine-readable and -executable content. All essential granular information units can be clearly identified and their reciprocal relationships recorded and made available for further processing or partial implementation.
- Level 4: Machine-interpretable content. The information in a standard is linked with implementation and usage information in such a way that it is implemented by machines directly or interpreted and combined with other information sources so that complex actions and decision-making processes take place automatically.

The German Institute for Standardisation (DIN) and German Association for Electrical, Electronic & Information Technologies (DKE) have presented an additional level that goes beyond Level 4 in the IEC model [64]:

- Level 5: Machine-controllable content. The content of a standard can be amended by machines working unassisted and adopted by automated (distributed) decision-making processes. The content adopted in this way is automatically reviewed and published via the publication channels of the standardization organizations.

The model emphasizes well the difference between machine-readable, -executable, -interpretable, and -controllable formats. Ideally, for the development of digital calibration management processes, the requirements would have to be machine-interpretable, enabling requirement-specific data and criteria such as allowance limits to be directly interpreted by the management systems as process parameters, making further automation of the processes management possible. Currently, when the requirements are not yet available in such a format, this kind of specific information must be included in the systems through human interpretation, which means that revisions of the requirements will require updating of the systems accordingly.

## 3. Materials and Methods

### 3.1. Studied Cases for the Proof of Concept (POC)

In the proof of concept (POC), the focus of the work was to harmonize the data exchange between the parties of a calibration chain to enable automation of the data management processes in the receiving end, thus eliminating the need for manual work to interpret and input the data into calibration management systems (CMS) or enterprise resource planning (ERP) systems, making the processes more efficient and improving the data integrity by reducing the chance of input errors. To achieve these goals, use of a multitenant platform was chosen as the basis for defining the data exchange processes and methods.

The DCC formats used in the POCs were the DCC XML schema versions 2.4.0 (https://www.ptb.de/dcc/v2.4.0/dcc.xsd, accessed 6 June 2022) in the Case 1 and 3.0.0-rc2 (https://www.ptb.de/dcc/v3.0.0-rc.2/dcc.xsd, accessed 6 June 2022) in the Case 2. In both cases, the means of using the DCC schema were agreed based on the requirements and needs of the participating organizations representing the calibration customer. The DCCs used in the testing were developed based on examples of existing calibration certificates.

#### 3.1.1. Case 1

In the studied case 1, the exchange and handling of DCCs was tested by mimicking a realistic calibration chain from a national measurement standard of an NMI to a working standard of a pharmaceutical manufacturing company. The main objectives for the POC were achieving an efficient and easy-to-use method for managing calibrations and the related data exchange between organizations. The physical quantities for which the harmonized file formats were agreed on within the consortium were temperature and pressure, as the working standard in question is capable of measuring both quantities. In addition to Aalto University, the organizations involved were:

1. VTT MIKES, the Finnish NMI;
2. Vaisala Oyj, an instrument manufacturer with an accredited laboratory and a provider of calibration services;
3. Orion Oyj, a pharmaceutical company;
4. Beamex Oy Ab, a field calibration instrument and CMS provider.

Figure 3 shows the POC setting, where different colors highlight the systems and operations of the organizations involved.

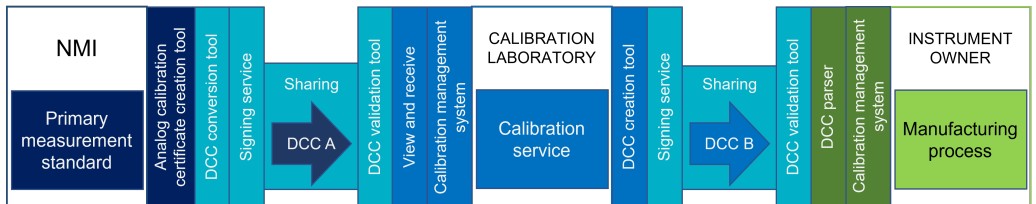

**Figure 3.** An illustration of the POC implementation. VTT MIKES is represented in dark blue, Vaisala in blue, Aalto University in turquoise, Beamex in dark green, and Orion in light green.

In case 1, a separate sub-schema was developed based on the DCC schema version 2.4.0 to limit the options for presenting the data only to those deemed necessary for the DCCs that were created as examples for testing.

### 3.1.2. Case 2

The main principles and objectives of the second studied case were mostly similar to case 1 as case 2 focused on the management of the DCCs of temperature measurement instruments and their transfer between the instrument owner and calibration provider. This work was carried out in alignment with the German Gemimeg II research project, in which the DCC-related development work has been led by PTB [5].

In addition to Aalto University, the organizations involved were:

1. PTB, the German NMI and developer of the DCC concept and XML schema;
2. Testo Industrial Services GmbH, a calibration service provider with an accredited laboratory;
3. Boehringer Ingelheim, a pharmaceutical company;
4. Beamex Oy Ab.

Based on the requirements defined in case 2, PTB developed a DCC good practice for temperature certificates based on the DCC XML schema version 3.0.0. The DCC good practice for temperature certificates has been further developed in the Gemimeg II project [65].

### 3.2. *Calibration and Data Management Processes*

The starting point for the development work was to assess the current practices and processes of the industry partners to observe the general requirements. Although there can be slight variations in the processes of different organizations, e.g., due to their size and the number of calibrated instruments, the main principles remain the same. Based on the processes of the industry partners, generalized overviews of the processes and workflows were then formed and utilized to define the requirements for the DCC-based processes and the POC platform implementation.

In general, calibrations are based on bilateral relationships between a service provider and an instrument owner, i.e., the customer. The process begins with a calibration order or request issued by the instrument owner. If the calibration is to be performed in an external laboratory, the instrument needs to be delivered as well. Upon receipt of the request and instrument, the calibration is performed and results are analyzed by the service provider based on the requirements defined by the customer. If an instrument does not perform within its tolerance, it will need to be adjusted or repaired, after which it must

be recalibrated. The calibrations before and after any significant changes made to the calibrated instrument are commonly referred as "as-found" and "as-left" calibrations, respectively [11]. To report the results of the calibration and a statement on the conformity of the instrument as required by the customer, the calibration vendor issues a calibration certificate, sends it to the instrument owner, and returns the instrument. At the receiving end, the calibration certificate is inspected to make sure it is correct and that it meets the requirements that have been defined for the calibration process. Once the certificate is deemed to fulfill the requirements, the data are imported into a calibration management system (CMS).

The calibration process should be considered as a closed-loop process between the instrument owner, i.e., calibration customer and calibration service provider. Thus, fully standardizing the communication and ensuring interoperability in wider industrial use requires also a format for the digital calibration request (DCR) in addition to the DCC. However, in the POC, the implementation concentrated only on the use of the DCC as its development has progressed further. Figure 4 shows a simple illustration of the communication between the instrument owner and calibration provider.

CALIBRATION ORDER

| CALIBRATION PROVIDER | | INSTRUMENT OWNER |
| --- | --- | --- |

CALIBRATION CERTIFICATE

**Figure 4.** An illustration of the closed-loop communication in a calibration process.

It is currently common that several different systems are used for the exchange and handling of the information in different parts of the process. An example of the systems used in the process is illustrated in Figure 5.

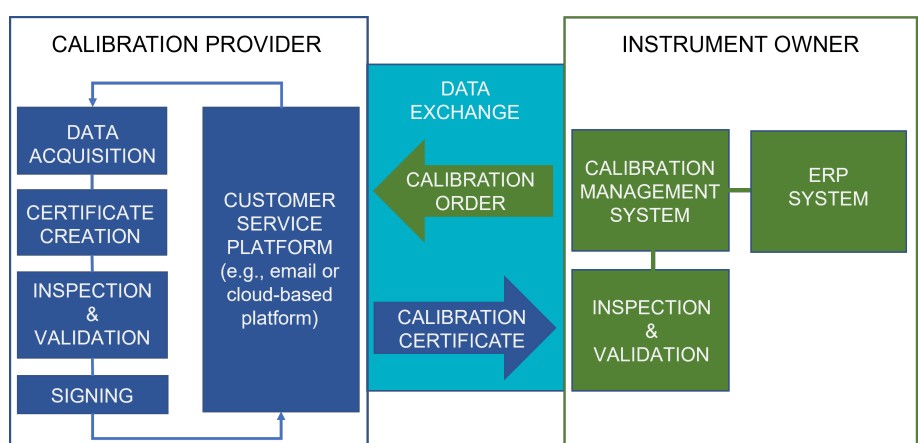

**Figure 5.** An illustration of the systems used in the calibration process and data exchange between the calibration provider and instrument owner.

### 3.3. Digitalizing Calibration Data Exchange and Management

Digitalizing calibration data management will enable the closed-loop process and communication without any manual conversions or loss of data between the systems, making the process robust in terms of data integrity. For the industrial companies with high numbers of annual calibrations, the benefits of the digitalization can be significant merely based on the reductions in the calibration data processing time and need of labor alone.

To digitalize the processes while maintaining the bilateral relations as close to their current state as possible, several organization-specific integrations would be needed between all the cooperators. To simplify the data exchange and management and improve

the efficiency of the development by concentrating the development load, an alternative approach was taken by focusing on the combination of several bilateral relations more as an ecosystem formed by the companies and their subcontractors and service providers. This does not affect the confidentiality of the information by default, but it enables the sharing of anonymized data confidentially with the partners involved, potentially providing shared benefits within the ecosystem.

### 3.3.1. Calibration Provider

As the sizes and calibration volumes of calibration service providers vary greatly, there are large differences in the systems used for creating calibration certificates. Currently, it is still common that especially the smaller calibration service providers' laboratories may still use quite simple solutions for creating their calibration certificates, such as basic text editor tools and templates where the information is filled in manually. For the larger organizations, the investments in more advanced IT systems and automation are easier to justify and more beneficial.

For the organizations using more advanced systems, adapting to a fully digital and machine-readable format is typically a lot less challenging as the data formats that have already been in use can, in most cases, be converted into a different with ease; for them, it is mostly simply a data-mapping task. For the organizations using more simple tools, the transition can be more challenging.

For data integrity reasons, the calibration certificates need to be signed when data are exchanged in a digital format, even if a signature is not considered to be a mandatory part of a calibration certificate, as is the case with ISO 17025. The reason behind this is that a digital signature can be validated to prove the authenticity and integrity of a document or file, i.e., ensure that it is authentic, originally created, and sent by the correct person or organization, and that its contents are unaltered.

The sharing of digital documents can be implemented in various ways; they can be sent via email, uploaded to a cloud server, or even delivered on a USB memory stick, to name a few possibilities. The differences in these methods are in the ways in which they can be integrated into the systems used to manage the documents.

### 3.3.2. Calibration Customer

On the industrial level, the calibration orders are typically based on a service contract that includes all the calibrations and services that the parties have agreed to be carried out over a defined period, e.g., a calendar year. In the manufacturing industry's end of the calibration chain, especially in non-accredited field calibrations, the calibration request has a significant role in the calibration processes as the requests or contracts include the instructions for the calibration procedures as mandated by the quality management regulations. For accredited laboratories, this is not a significant issue as their procedures are audited as a part of the accreditation process, based on which they are also compliant with the quality system regulations. In the field calibration cases, it is mostly up to the manufacturing companies to audit the service providers that they use and ensure that the calibrations are performed correctly.

The correct interpretation of the exchanged data in this kind of setting is heavily dependent on the format in which the data are exchanged and their harmonized use by the parties involved. This is why it is necessary for the receiver to have the means to validate that the received certificate uses the correct format in a correct manner, to make sure that the data can be interpreted and used correctly when they are uploaded to the CMS. Due to the use of digital signatures, validation also includes authenticating the source of the certificate and checking that it is unaltered to ensure its data integrity.

Long-term storing and archiving of calibration certificates is required for investigating any measurement-related quality issues and for auditing purposes. For electronic records and documents, the storage can be implemented with relative ease with conventional cloud storage applications. In addition to maintaining the records over the required period, they

also need to be easily accessible and viewable, especially for auditing purposes, which can be implemented in the calibration management systems with relative ease.

The availability of calibration data in a machine-interpretable and processable format is one of the cornerstones for the use of the data in Industry 4.0 applications. For example, the data could be used to analyze the performance of individual devices, device models, or types over time to predict the optimal calibration intervals and select the devices used for controlling and processing the manufacturing processes. These kinds of features have already been available in commercial calibration management systems that have been developed for use in the industry, where field calibrations are common. However, the use of the features is limited to certificate formats used by collaborating instrument and system providers, meaning that a significant amount of the data is not usable without manually importing the data, which is an issue in terms of data integrity.

### 3.4. Aims and Requirements for the POC DCC Platform

Ideally, the platform could be used through APIs, allowing the processes to be highly automated. However, due to the stage of the development work and the variances in the uptake capabilities of organizations, it is likely that there will be a lengthy transition period in which digitalization within the calibration ecosystem will progress step-by-step. This is why some of the features of the platform were developed to be used both manually and with an application programming interface (API). The aims of the POC were to make the different steps of the DCC exchange easy and efficient while also fulfilling the requirements of data integrity applied in the current calibration management procedures. Table 2 represents features that were considered necessary for the testing of the platform based on the defined aims and requirements.

**Table 2.** The necessary features for the platform based on the defined aims and requirements.

| | |
|---|---|
| Universal features | User management based on organizational user credentials<br>Uploading and downloading of DCCs<br>A list view of DCCs created by or shared with the user<br>An easy-to-use user interface (UI) to enable manual use of the other features |
| Features necessary for a calibration provider | DCC creation using an input form type of a tool, for cases where a more advanced DCC creation is not available<br>DCC signing<br>DCC sharing |
| Features necessary for a calibration customer | A notification when a DCC is received<br>A human-readable format for viewing DCCs<br>Signature validation |

## 4. Results

### 4.1. Data Transfer Utilizing the POC Platform

The features that were deemed necessary for the exchange of DCCs were implemented using five application programming interfaces (APIs):

1. A signing API for creating digital seals for DCCs;
2. An import API for uploading DCCs to the DCC Platform;
3. A sharing API for granting access rights to DCCs;
4. An export API for retrieving DCCs from the DCC Platform; and
5. A validation API for validating the digital seals of received DCCs.

The data exchange process based on the APIs is presented in Figure 6.

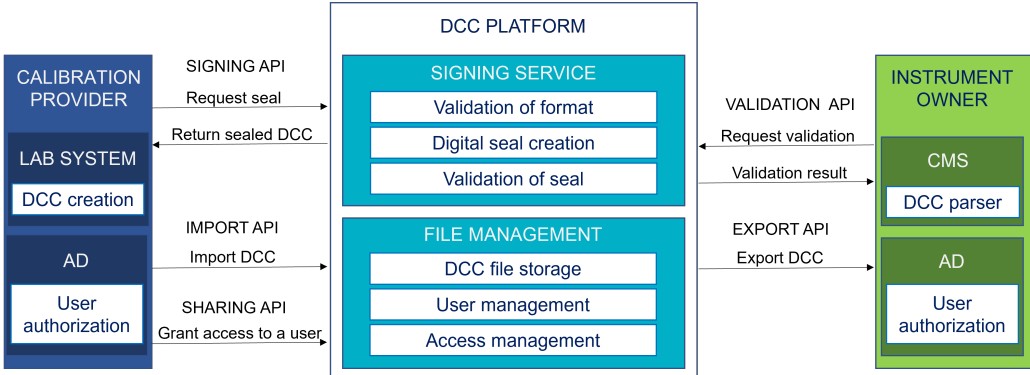

**Figure 6.** An illustration of the DCC Platform-based exchange of DCCs using the APIs of the platform.

To keep the platform simple, easy to use and deploy, and cost-efficient, existing and commonly used technologies were intended to be utilized. Consequently, the file management and storage of DCCs on the platform was implemented based on Nextcloud (https://nextcloud.com/about/, accessed 6 June 2022) and the supporting features such as the user interface (UI) and access management were developed to enable better compatibility for the requirements of the use cases.

Respectively, the securing of the confidentiality of the data exchange on the platform through user management and authorization was implemented using Azure Active Directory (AD) (https://azure.microsoft.com/en-us/services/active-directory/, accessed 6 June 2022), which is a widely utilized user identity management system also used by the participating organizations.

*4.2. Features for a Calibration Provider*

The creation tool for DCCs was implemented in two ways. The first one was a conversion tool that was added to an existing in-house-developed calibration certificate creation software that allowed the creation of a DCC in a similar way to the conventional certificate. The second method was by using a simple data input form that was developed for manually creating DCCs.

The signing of the documents was implemented using organizational seals, which are digital signatures that identify the organization authenticating the document instead of the person creating it. In the POCs, the identifications of the persons who performed the calibration were stated in the DCC. The organizational seals are well suited for organizations that already have established personnel authentication methods that can be applied to authorize the rights to seal DCCs. As the rights are managed based on the existing authentication system, the issuing or removing of the signing rights happens automatically as a part of organization's normal onboarding and offboarding processes. Depending on the system used by the calibration provider, two versions for sealing were implemented:

- A physical signing server using an Intel NUC mini PC, allowing the sealing to be done locally and separated from the cloud platform;
- A cloud-based sealing tool as a part of the platform, enabling the sealing and sharing of the document with the same user interface.

The seals for the DCCs were created and validated using public and private key pairs designated to each organization issuing the DCCs in the POCs. For the authorization of the keys, a simple prototype PKI was set up at Aalto University, with the root key being securely stored in a safe. The keys issued for the testing were safely either stored in the NUCs or in the cloud depending on which system the organization was using. In either case, the authorization of key use was managed by the corresponding organization. For testing purposes, also separate, universally usable test users were created to enable using the platform without individual AD credentials.

To ensure that the format of the DCC was following the agreed structure, the DCC was validated against the DCC schema or good practice schema before a seal was added to the file. To further ensure that the information was interpretable, the files were canonicalized as a part of the seal creation process to remove non-meaningful variances in the documents. The signing service also included a function for validating the seals.

### 4.3. Features for a Calibration Customer

In the studied cases, the validation of the data format was performed when the calibration provider uploaded the DCC to the DCC platform. For this reason, one key benefit of the platform is that it ensures the interoperability between the platform users as all the data are validated when imported to the platform. Using the signing service, the instrument owner can validate the digital signatures to the received DCCs to ensure that the received data are authorized by the expected organization and that the document has not been manipulated.

After the data integrity of the document is validated, the calibration data need to be imported to the calibration customer's CMS. For this purpose, an API was developed to enable integration between the DCC platform and receiving CMS. In the receiving CMS, a parser was used to convert the calibration data within the DCC to the data format used in the CMS. This allowed the imported calibration results to be reviewed and approved in the system using existing features to return the instrument for production use.

Alternatively, the human-readable format of the DCCs also allowed the data to be imported manually, similarly to how it is performed with paper or PDF certificates. However, this would also entail data integrity issues caused by the human interaction, making the approach far from ideal in comparison to the automated process.

### 4.4. Testing and Validation of the Platform-Based Calibration Data Exchange and Management Processes

The operation of the features of the platform defined in Table 2 was examined based on the sufficiency of the implemented functionalities per feature and the technological readiness of the features. The results of the testing and validation are presented in Table 3. The testing indicated that the platform met its requirements and was mostly working as intended. Some of the features would benefit from further development and refinement. This was expected since the platform was developed as a POC and not a finalized system for industrial use as such.

**Table 3.** Testing results of the features defined in Table 2. The functionality and technological readiness were evaluated as follows: 2 = pass, usable as is; 1 = pass, usable but improvable; 0 = fail.

| Universal Features | Functionality Readiness | Technological Readiness |
| --- | --- | --- |
| User management with Azure AD | 2 | 2 |
| DCC import and export APIs | 2 | 2 |
| UI for manual use | 1 [1] | 2 |
| Listview for DCCs | 2 | 1 [2] |
| Features necessary for calibration provider | | |
| DCC input form | 1 [3] | 2 |
| DCC sealing | 2 | 2 |
| DCC sharing | 2 | 1 [4] |

**Table 3.** *Cont.*

| Universal Features | Functionality Readiness | Technological Readiness |
|---|---|---|
| Features necessary for calibration customer | | |
| Notification upon DCC receival | 1 [5] | 1 [5] |
| Human-readable format for DCCs | 2 | 2 |
| Seal validation | 2 | 2 |

[1] The implemented functionalities of the UI were technically sufficient but additional functionalities would make it more user-friendly. [2] The required functionalities worked as intended, but the user experience was not refined within the POC. [3] Input form did not include an option to preview DCCs or import calibration results as a file. Otherwise, the implemented features worked as intended. [4] Sharing worked otherwise as intended, but data ownership was not transferred to the receiver as a part of the sharing process. This mainly affects the long time storing and management of the DCCs. [5] Notifications were implemented based on the Nextcloud notification system, which could not redirect the user to the DCC platform UI.

## 5. Discussion

In the testing and validation, the POC implementation of the DCC-based calibration data exchange and management platform proved to work as intended, providing an efficient way to exchange calibration data digitally, as all the features fulfilled the requirements. As the implementation was only developed as a POC, some areas for additional refinements were also found, which was as expected.

Due to the work being the first collaboration initiative for POCs on the utilization of DCCs for the participating organizations, the scope of the work was kept relatively narrow in terms of the whole calibration management process. Consequently, the results of the POC do not fully cover all parts of the processes that are necessary in the industrial calibration management. Still, the work provided valuable knowledge and a basis for further collaborative development in digitalization.

The chosen approach for implementing the platform was developed based on a relatively simple infrastructure utilizing already widely used services, such as the Azure AD. In addition, the signing and validation service was integrated into the platform. Thus, the number of new and separate systems required for enabling the secure use of DCCs was minimized. This approach suits well the requirements of the industry, especially when smaller organizations are considered, as simplicity and ease of deployment were given relatively high priority. For example, in the testing and deployment of the platform, the cloud-based implementation of the signing service was found to be the preferred option compared to the server being deployed at the corresponding organization. In the tested cases, the data exchange was limited to only DCCs, but the same approach will be applicable also for other similar data formats or documents, e.g., calibration orders or invoices.

However, there are limitations for implementing the infrastructure in such a way for the NMIs and other organizations essential for maintaining national and regional infrastructures, as, e.g., dependencies on services provided by foreign companies may be considered as a threat to the security of the infrastructure. For these situations, approaches based on the proprietary infrastructures, such as the approach used on the Metrology Cloud [13], could be the better alternative, despite the potential complexity or significantly higher operation costs. Naturally, different types of solutions will be ideal for different organizations; thus, the availability of both simple and robust implementations would be beneficial provided that the compatibility and interoperability of the different systems can be ensured.

### 5.1. Challenges in Advancing the Digitalization of the Metrology Infrastructure

Although there are several ongoing efforts for advancing digitalization in metrology, from a broader view, the work is still at an early stage. Effectively, it will take several years before the transition of the whole MI into a digital environment is complete. Until then, the benefits of the digitalization will only be partial. However, even these partial benefits can

be significant for the organizations, if closely related organizations working in the same domains are willing to collaborate.

The limitations for the uptake of digital solutions vary between the organizations within the metrology infrastructure based on their roles. The direct benefits of the digitalization are more significant the larger the amount of measurement devices and instruments used by the organization, which of course leads to the manufacturing companies obtaining the most benefits. From their point of view, the main concerns are justifying investments in digital solutions in terms of the stance of the regulation bodies towards the developed technologies. This is why active collaboration between the solution developers and regulators is necessary.

For the smaller organizations within the metrology infrastructure, e.g., calibration laboratories and service providers specialized in a specific domain, the direct benefits of the digitalization may not be significant enough that the investments for new systems could be justified by improvements, e.g., in data management efficiency, alone. For these organizations, the motivation for the digital transition is based on the needs and demands of the customers and their willingness to contribute the additional costs from the required investments.

At the moment, the requirements defined for the quality management systems are defined in such a way that there are some flexibilities for implementing the systems digitally. This is mostly because the standards have been developed to be interpreted by humans and the organizations pursuing compliance with the requirements are obliged to prove that the systems developed based on their interpretation are fulfilling these requirements. This raises a challenge when the quality management systems are being digitalized as this kind of flexibility is not suitable for machine interpretation. Transition to digital systems means that the interoperability of the data formats and harmonization of the data management procedures should be given higher priority in the standardization and regulation work. A good example of how the collaboration of the technology developers and regulation bodies can be established is the participation of the German authorities in the Gemimeg II project, which is a national project where solutions for the use of the DCC are being developed based on the needs of the German industry. Ideally, the future development initiatives and involvement of regulation bodies should be expanded towards a global scale.

Fortunately, the slow progress of the digitalization of the metrology infrastructure also means that many of these challenges have been already solved in other applications, so there is already plenty of expertise and tools available. What is needed is that the experts of the areas of metrology and information technology come together to share this expertise and discuss the metrology-specific challenges that remain to be solved. In other applications, this kind of work has been made possible by foundations such as the Open Platform Communications (OPC) Foundation, where organizations have been able to actively participate and contribute to the development of standards and tools on behalf of the community based on their own specific areas of interest and expertise. This way, the development load can be efficiently shared within the community, and the situations in which different consortia would be working on the same topics totally unaware of each other can be minimized.

### 5.2. Future Work and Research Topics

The strict regulations on data integrity in the pharmaceutical industry have lead to significant interest in digitalization. Thus, eagerness for the early adoption of DCCs has also been high. Consequently, advancing DCC development requires extending the work and tests to other domains. Ideally, the testing should result in the development of good practices similar to the one developed for temperature calibrations in the presented case 2.

As calibration, similarly to other maintenance operations, is a necessity for manufacturing companies for compliance reasons, it cannot truly be considered as a major area of competition for the companies but more of a common obstacle. This makes the joint development of calibration management-related technologies and sharing of knowledge a perfect example of anti-rival good, which is where the concept of a joint platform for the

calibration ecosystem as an enabler for, e.g., big data, becomes prominent. For example, sharing some of the information about, e.g., instrument performance and calibration history in certain process conditions allows the companies to better understand the behavior of different instrument models or types, enabling more precise and optimized selections for the monitoring and controlling of processes. Respectively, the instrument manufacturers and calibration service providers could also receive a lot of valuable information about the usage and instrumentation needs in their current and potential customer bases, allowing them to produce and offer products and services more suitable for the needs of the customers.

In addition to the possibilities in improving the calibration data management of the instrument owners, the digital transition could be harnessed to make the audition and accreditation procedures more streamlined. For example, instead of inspecting the calibration information of randomly selected individual devices of a process line, the information could be made available for separate auditor interfaces, where the collective information of the measuring equipment of an entire process line could be made available for the auditors to access and view so that the audition could, at least by those parts, be possible to perform remotely in real time. Investments in the auditing infrastructure could be beneficial in lowering the costs of these procedures and help to increase the number of parties covered by accreditation, thus improving the overall quality and reliability of the operations within the metrology infrastructure.

## 6. Conclusions

Calibrations are an essential part of quality management in industries where measurements are required for monitoring and controlling processes. The recent developments in the digitalization of the global QI and metrology are enabling a transition towards automated calibration management processes. In addition to improved efficiency, another key benefit of digitalized calibration data management is improved data integrity, which has been one of the main interests in the development of IoT systems in the pharmaceutical industry. However, the digitalization of processes involving data exchange between several organizations can only be managed through standardization of the used data formats and interfaces. Due to the bilateral nature of calibration services, the service and system providers have been conservative in relation to collaboration, which has made further standardization difficult. To effectively advance the digitalization of calibration management, this kind of mindset would need to change from strictly bilateral relations towards a joint ecosystem.

In this paper, we presented a concept of a multitenant platform providing the necessary interfaces and functionalities for the secure exchange of DCCs as a method for ensuring interoperability between organizations. For large companies in the industry that use merely the improvements in working efficiency can provide significant cost savings, which would justify the necessary investments in digitalization. However, the full benefits of the digitalization can only be achieved if even the smallest service providers are able to adapt to the necessary changes. For them, justifying any investments may not be possible in terms of efficiency alone. This is an issue where the platform approach for organizing the collaboration between the organizations can help, as the necessary digital operations could be provided by a central service provider, collectively reducing the development work within the ecosystem.

The testing performed based on real use cases in the pharmaceutical industry proved that the multitenant platform is an efficient means of organizing the exchange of DCCs. Considering the strict regulations of the pharmaceutical industry, the results indicate that the system would suit the requirements of other industries as well, providing cost-efficiency through economies of scale. However, the optimal means of implementing such a system can vary greatly with each organization. As stronger security solutions are more costly to develop and operate, a thorough assessment of potential risks and threats is essential for determining the level of security required. For example, the requirements of the industry

are typically not as high as the requirements of the national infrastructures essential for society, in which case more advanced and robust solutions are ideal.

**Author Contributions:** Conceptualization, T.M., J.N. and R.V.; methodology, T.M. and J.N.; validation, T.M. and J.N.; investigation, T.M.; writing—original draft preparation, T.M.; writing—review and editing, T.M., J.N. and R.V.; visualization, T.M.; supervision, R.V. All authors have read and agreed to the published version of the manuscript.

**Funding:** This work was funded by Business Finland, grant number 1811/31/2019, in association with Aalto University.

**Institutional Review Board Statement:** Not applicable.

**Informed Consent Statement:** Not applicable.

**Data Availability Statement:** Not applicable.

**Acknowledgments:** The authors would like to thank all the team members and participating organizations in the POC for their help and contributions.

**Conflicts of Interest:** The authors declare no conflict of interest.

## Abbreviations

The following abbreviations are used in this manuscript:

| | |
|---|---|
| AD | Active directory |
| API | Application programming interface |
| CMS | Calibration management system |
| DCC | Digital calibration certificate |
| DI | Designated institute |
| DIN | German Institute for Standardisation |
| DKE | German Association for Electrical, Electronic & Information Technologies |
| ERP | Enterprise resource planning |
| EMPIR | European Metrology Programme for Innovation and Research |
| FDA | US Food and Drug Administration |
| GAMP | Good advanced manufacturing practice |
| GLP | Good laboratory practice |
| GMP | Good manufacturing practice |
| GXP | Good practice |
| IEC | International Electrotechnical Commission |
| INetQI | International Network on Quality Infrastructure |
| ISA | International Society of Automation |
| ISPE | International Society of Pharmaceutical Engineering |
| MHRA | Medicines and Healthcare products Regulatory Agency |
| MI | Metrology infrastructure |
| NMI | National metrology institute |
| OPC | Open Platform Communications |
| PDF | Portable document format |
| POC | Proof of concept |
| QI | Quality infrastructure |
| RMO | Regional metrology organization |
| SIM | Inter-American Metrology System |
| TQM | Total quality management |
| UI | User interface |
| XML | Extensible markup language |

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
