# Peer review of "Digitalization of Calibration Data Management in Pharmaceutical Industry Using a Multitenant Platform"

_applsci, doi:10.3390/app12157531_

Round 1

Reviewer 1 Report

This paper investigates the general requirements for calibration data management and procedures as a part of quality management in the pharmaceutical industry.

Introduction: A few more requirements for the pharmaceutical industries pertaining to quality control need to be included.

The paper designed is poor and it seams more of report writing instead of a research paper.

My suggestion is to redegine the paper and put more emphasis on the actual contribution of authors.

Most of the literature is explained with bullet point which seam odd in research articles.

The discussion part does not provide any comparison or relevance with previous study. I have a doubt whether is it a first study on this kind?

Further, conclusion can be improved by restating the problem and than put stress on actual funding of the study.

Reviewer 2 Report

The theme of this paper is interesting and innovative, and certainly useful.

This paper could be very interesting for researchers looking for this Journal.

However, this paper has some weaknesses.

ABSTRACT

the abstract is what attracts (or does not) the attention and interest to the article. So, it should be carefully written.

It is suggested that they simplify the text in relation to the methodology.

1.INTRODUCTION

An introduction should be informative and well-worded. Therefore, it is missing:

-     present the theoretical problem/formulation and the objective of the study.

-     give clues to the discussion of the results.

3. MATERIALS & METHODS

It shall provide the necessary and sufficient information to assess how the study was conducted in order to allow its reproduction by other.

It is necessary to present as limitations of the applied methodology

4. RESULTS

This point should be articulated with the previous one, as the results presented should be supported by the descriptive methods.

The results should show the evidence of the study and should be presented according to a logical and informative and perceptible sequence.

5. DISCUSSION

More than describing the processes and procedures, it is necessary to reflect and discuss the results obtained.

This point should be improved.

CONCLUSION 

The results should be presented in a comprehensive manner, highlighting the most relevant ones and do not provide a summary of the text.

The way in which the results relate to each other and with those of other studies should be evidenced.

The future investigation need referred.

The theoretical implications and possible practical applications should be discussed.

The originality and relevance of the results presented should be strengthened.

The limitations of the study itself should be presented and discussed.

None of this has been presented at this point.

Round 2

Reviewer 2 Report

The article is now ready to be published